# Reaction engineering blocks ether cleavage for synthesizing chiral cyclic hemiacetals catalyzed by unspecific peroxygenase

Xiaofeng Han[1,2,5], Fuqiang Chen[2,5], Huanhuan Li[2,3], Ran Ge[2], Qianqian Shen[2,3], Peigao Duan [3] ✉, Xiang Sheng [2,4] ✉ & Wuyuan Zhang [2,4] ✉

Hemiacetal compounds are valuable building blocks in synthetic chemistry, but their enzymatic synthesis is limited and often hindered by the instability of hemiacetals in aqueous environments. Here, we show that this challenge can be addressed through reaction engineering by using immobilized peroxygenase from *Agrocybe aegerita* (*Aae*UPO) under neat reaction conditions, which allows for the selective C-H bond oxyfunctionalization of environmentally significant cyclic ethers to cyclic hemiacetals. A wide range of chiral cyclic hemiacetal products are prepared in >99% enantiomeric excess and 95170 turnover numbers of *Aae*UPO. Furthermore, by changing the reaction medium from pure organic solvent to alkaline aqueous conditions, cyclic hemiacetals are in situ transformed into lactones. Lactams are obtained under the applied conditions, albeit with low enzyme activity. These findings showcase the synthetic potential of *Aae*UPO and offer a practical enzymatic approach to produce chiral cyclic hemiacetals through C-H oxyfunctionalization under mild conditions.

Selective functionalization of carbon–hydrogen (C-H) bonds is a powerful strategy in installing new functional groups into organic molecules, whereby the synthetic routes are potentially largely simplified (e.g., avoiding tedious and costly prefunctionalization steps) for more sustainable chemical synthesis[1,2]. Over the last decade, organic chemistry has witnessed significant advances in transition metal-based complexes[3–5] and organocatalysts[6] used for C-H bond functionalization reactions. On the other hand, enzymes are the method of choice for catalyzing challenging reactions such as C-H bond activation and functionalization[7–9]. When a metal ion or organic cofactor is embedded into the well-evolved supermolecular structure of a protein, the regio- and stereoselectivity are granted in a catalytic process. In particular, oxidoreductases such as oxidases, monooxygenases, and peroxygenases have been widely investigated for selective C-H functionalization[7–12]. Using the technology of directed evolution, C-H

functionalization can be expanded to non-natural reactions such as transforming C-H bonds into C-Si[13], C-B[14], and C-N[15] bonds. Among these enzymes, heme-thiolate-containing enzymes such as P450 monooxygenases and unspecific peroxygenases (EC 1.11.2.1, UPOs) relying on oxoferryl-heme as the oxygenating species (so-called compound I) have been investigated thoroughly for C-H bond oxyfunctionalization reactions[12,16]. P450s are typically dependent on nicotinamide cofactors (NADP(H)) coupled with a regeneration system to reduce $O_2$ as the oxygen source, and the resultant sophisticated electron transport chains impose low efficiencies on the catalytic transformations. In contrast, the UPOs directly use partially reduced $H_2O_2$ as an oxygen- and electron-source to catalyze C-H bond oxyfunctionalization. In principle, UPOs have substrate and product scopes similar to those of P450s but with higher efficiencies (e.g., higher turnover numbers)[12].

[1]College of Chemistry and Materials Science, Inner Mongolia Minzu University, Tongliao 028000, China. [2]Tianjin Institute of Industrial Biotechnology, Chinese Academy of Sciences, 32 West 7th Avenue, Tianjin 300308, China. [3]School of Chemical Engineering and Technology, Xi'an Jiaotong University, Xi'an 710049, China. [4]National Center of Technology Innovation for Synthetic Biology, 32 West 7th Avenue, Tianjin 300308, China. [5]These authors contributed equally: Xiaofeng Han, Fuqiang Chen. ✉e-mail: pgduan@xjtu.edu.cn; shengx@tib.cas.cn; zhangwy@tib.cas.cn

To date, the UPOs have shown promiscuous activity and enable a wide variety of reactions, such as hydroxylation, epoxidation, demethylation, halogenation, sulfoxidation, and alcohol oxidation[17]. In particular, UPOs-catalyzed hydroxylation could accept benzylic, aromatic, allylic or propargylic C-H bonds in a rather regio- and/or stereo-selective manner, giving access to various precursors[18] and drug analogs that are industrially relevant[19]. Very recently, we also demonstrated that peroxygenase enabled shifting the reactivity of aromatic C-H hydroxylation towards a dearomatization reaction by capturing the aromatic oxide, an instant intermediate for the NIH shift to phenols[20].

Of note, the established demethylation is a less explored reaction for UPOs, which convert ethers to cleaved products via a hydrogen abstraction of the alpha C-H bond and an oxygen rebound mechanism[21,22]. The demethylation reaction thus far, as suggested by its name, only resulted in the formation of cleaved products due to the instability of the hemiacetals towards spontaneous hydrolysis in aqueous conditions (e.g., sole formation of 4-hydroxybutanal via tetrahydrofuran oxidation shown in Fig. 1a). Interestingly, peroxygenases have shown unusual stability in pure organic solvents upon immobilization[23,24]. This inspired us to assume that the hydrolysis of hemiacetals obtained from immobilized peroxygenases can be circumvented under neat reaction conditions. Hemiacetal compounds are valuable synthetic intermediates in chemical synthesis[25]. For example, cyclic hemiacetals frequently act as vital functional moieties in pharmaceutical development (Supplementary Fig. 1). Traditionally, the synthesis of hemiacetal compounds is primarily restricted to the addition reaction between an alcohol and an aldehyde/ketone, or obtained by reducing lactone with metal catalysts[26,27]. However, hemiacetals have not been deemed significant for UPOs or enzymes in general due to their instability in aqueous conditions[28].

Herein, we demonstrate the enzymatic synthesis of cyclic hemiacetals by peroxygenase-catalyzed C-H bond oxyfunctionalization under neat reaction conditions (Fig. 1b). The reaction engineering allowed us to switch the peroxygenase-catalyzed ether cleavage to hemiacetals synthesis. This work will not only expand the scope of the known C-H oxyfunctionalization chemistry enabled by peroxygenases but also contribute a catalytic method for hemiacetal synthesis in the synthetic community.

## Results and discussion
### Proof-of-concept experiments
We began to evaluate the envisioned hemiacetal synthesis by using UPO under neat reaction conditions. The hydroxylation of tetrahydrofuran (THF, **1**) to tetrahydrofuran-2-ol (**1a**) was chosen as a model reaction (Fig. 2a). THF serves a dual role as both the substrate and reaction medium. The recombinant peroxygenase from *Agrocybe aegerita* (r*Aae*UPO, PaDa-I variant) was prepared according to a previous protocol[29]. To obtain immobilized enzyme, we first screened seven commercial resin carriers that have abundant amino groups (LX 700, 703, 704), epoxy groups (LX 600, 603, 609) or strong adsorption capacity (LX 1000). The amino resin was activated with glutaraldehyde

before immobilization, while the other resins were used without any pretreatment. Although a number of oxidative in situ generations of $H_2O_2$ have been reported for UPOs[30–37], we decided to use a syringe pump for the supply of $H_2O_2$ to the enzyme to reduce the complexity of the overall reaction system. Under the reaction conditions chosen, the amino resin LX 700 excelled over the other six in terms of the desired product concentration (Fig. 2b) and the amount of the enzymes immobilized (Supplementary Table 1, Supplementary Fig. 2). A steady increase in product formation was observed, with a significant amount of **1a** (24.6 mM) obtained in 24 h. 2.8 mM of the overoxidized product γ-butyrolactone (**1b**) was observed (Fig. 2c). The formation of products **1a** and **1b** is contradictory to the established THF oxidation catalyzed by peroxygenase, in which only hydroxylated 4-hydroxybutanal was observed under aqueous condition[21]. We attribute this effect to the improved stability of cyclic hemiacetal product **1a** under near-neat reaction conditions.

### Characterization of the enzymatic oxyfunctionalization reaction
The robustness of the synthesis of **1a** was determined by the performance of the immobilized peroxygenase, the combinatorial supply of $H_2O_2$ and its consumption by the enzyme under neat reaction conditions. In a next step, we systematically varied these parameters to optimize the reaction conditions by using the r*Aae*UPO immobilized on LX 700 (Table 1). In terms of the enzyme concentration, a value of approximately 100 mg (corresponding to 1.45 µM) was found to be optimal with respect to the desired product concentration and the initial reaction rate of THF hydroxylation (Table 1, entries 1–3). This observation is reasonable, as at lower loading, the enzymes were inactivated by $H_2O_2$ feeding with an applied rate of 6 mM h$^{-1}$, while at higher enzyme loading, a mass transfer limitation could slow down the reactions (Supplementary Fig. 3). When the varied $H_2O_2$ was investigated, it turned out that at a rate of 6 mM h$^{-1}$ yielded the highest production formation (Table 1, entries 2, 4, and 5). Lowering the feeding rate to 3 mM h$^{-1}$ resulted in an almost linear increase in the product concentration. In contrast, a higher feeding rate could inactivate the enzymes quickly (Supplementary Fig. 4). With the experiments performed, only a minor concentration of the overoxidized lactone **1b** was observed, corresponding to an overall selectivity of approximately 90% (Table 1). A turnover number (TON) of 27380 was achieved for **1a** synthesis, which is superior to the natural enzyme activity of peroxygenases in C-H oxyfunctionalization reactions[38]. It is of interest to note that the efficiency of immobilized r*Aae*UPO in using $H_2O_2$ is significantly higher than that of $^t$BuOOH in terms of product concentration and selectivity (Table 1, entries 2, 6 and 7, and Supplementary Fig. 5). This is rationalized that the kinetically less effective $^t$BuOOH has a good affinity with r*Aae*UPO in apolar solvents, whereas $H_2O_2$ is expected to have a higher affinity with the enzyme in slightly polar solvent (i.e. THF)[39]. Additionally, a possible hydroxylation of the C-H bond leading to tetrahydrofuran-3-ol was not observed in all experiments as judged by gas chromatography using a commercial standard compound, suggesting excellent regioselectivity of the hydroxylation of THF by r*Aae*UPO. The control reactions using r*Aae*UPO or $H_2O_2$ alone, or using the resin and $H_2O_2$ together did not yield any oxidized products of THF (Table 1, entries 8–10).

### Substrate scope of the enzymatic cyclic hemiacetal synthesis
To show the applicability of the envisioned enzymatic approach for hemiacetal synthesis, we next investigated a range of cyclic ether substrates with electronic and structural diversity (Fig. 3, and Supplementary Figs. 6–18). The six- and seven-membered cyclic ethers were readily oxidized into the corresponding hemiacetal products (Fig. 3, **2a** and **3a**), while the conversion of oxetane was not observed. The highest enzyme activity with a TON of 95172 was achieved with tetrahydropyran (THP), which is comparable to the activity of

**Fig. 1 | Peroxygenase-catalyzed conversion of ether compounds. a** The established cleavage of ethers and **b** the proposed oxyfunctionalization for cyclic hemiacetal synthesis.

peroxygenases in cycloalkane oxidation[35,36,40]. 1,4-Dioxane and benzo-1,4-dioxane were solely converted into hemiacetals with lower enzyme activity (Fig. 3, **5a, 12a**). The substituted THF and THP were also accepted by r*Aae*UPO (Fig. 3, **6a-11a**). In particular, the dihydrobenzo-furan and dihydrobenzo-pyran compounds, which are potentially more interesting in the pharmaceutical industry, were oxidized into the desired hemiacetals albeit with varied enzyme activities. The synthesis of some cyclic hemiacetals was also performed on a pre-parative scale at 50 mL. The solvent was evaporated and recycled, and the products were purified via flash chromatography after 24 hours of oxyfunctionalization reactions. Isolated quantities of 0.28, 0.69, 0.21 and 0.51 g were obtained for **1a, 2a, 5a** and **10a** (Supplementary Fig. 19), respectively. The substrates bearing moderate deactivating groups (Fig. 3, **13a-16a**) were not converted. Additionally, a few non-cyclic ethers were tested (Supplementary Table 2) and only C-O bond cleaved products were obtained. Interestingly, although the hydro-xylation products of the two piperidine derivatives were not observed (Fig. 3, **17a** and **18a**), the corresponding lactam products via possible overoxidation of hemiaminals were obtained with 4.3 and 2.1 mM (Supplementary Figs. 17 and 18), respectively. This could be due to the low reactivity of r*Aae*UPO with piperidine compounds and the poor stability of hemiaminals[41].

Meanwhile, we also determined the enantiomeric excess (*ee*) of the obtained hemiacetal products. To determine the absolute configuration, (*R*)-selective alcohol dehydrogenase from *Lactobacillus kefir* was used to prepare the standard compounds with com-plementary chirality starting from lactones, respectively (Supplemen-tary Information). As shown in Fig. 3, while the model substrate did not show a chirality, >99% *ee* in (*R*)-configuration was achieved with **2a, 3a, 5a, 7a, 8a, 10a, 11a** and **12a** (Supplementary Figs. 28–36).

## Synthesis of lactones and lactams

Although a minor amount of lactones were formed along with the hemiacetal products, we wondered whether it is possible to switch the product distribution towards the formation of lactones by carrying out the reaction engineering, i.e. switching to an aqueous condition. The lactone compounds are versatile building blocks in flavors, fragrances, polymers, etc[42]. The enzymatic synthesis of lactones represents a green alternative to current chemical methods[43]. To show the enzy-matic synthesis of lactones starting directly from the C-H bond hydroxylation of cyclic ethers via the in situ use of hemiacetal (Fig. 4a), we then investigated the reaction pH, a crucial parameter determining the stability of the hemiacetals. As shown in Fig. 4b, a change of the pH

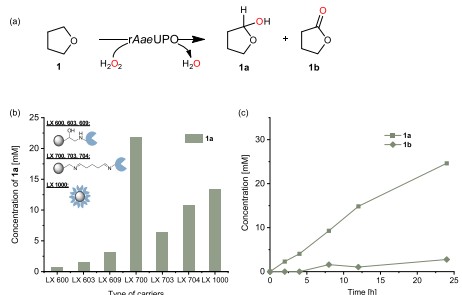

**Fig. 2 | Peroxygenase-catalyzed hydroxylation of tetrahydrofuran. a** Reaction scheme. **b** Screening of immobilized enzymes under neat conditions. Conditions: [THF] = 0.5 mL (12.3 M), [immobilized r*Aae*UPO] = 100 mg (corresponding to 1.45 µM), [H$_2$O$_2$] = 6 mM h$^{-1}$, 30 °C, 7 h. **c** Time course of the hydroxylation of tet-rahydrofuran yielding tetrahydrofuran-2-ol (squares, **1a**) and γ-butyrolactone (diamonds, **1b**). Conditions: [THF] = 0.5 mL (12.3 M), [immobilized r*Aae*UPO] = 125 mg (corresponding to 1.81 µM), [H$_2$O$_2$] = 6 mM h$^{-1}$, 30 °C, 24 h. The reported value is based on the mean value of two distinct experiments (*n* = 2).

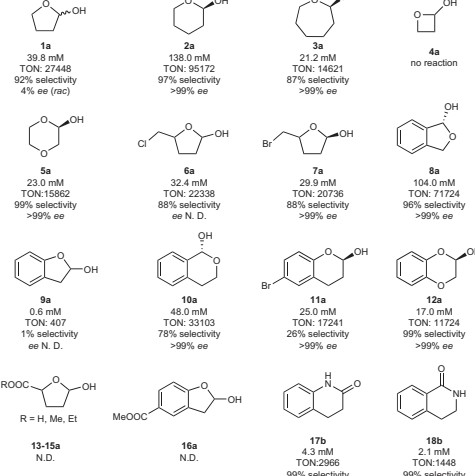

**Fig. 3 | Substrate scope of cyclic ether oxidation reaction.** Conditions: [sub-strates] = 0.5 mL, [immobilized r*Aae*UPO] = 100 mg (corresponding to 1.45 µM), [H$_2$O$_2$] = 6 mM h$^{-1}$, 30 °C, 24 h. The selectivity was determined by GC. Selectivity = [**1-18a**]/([**1-18a**]+[**1-18b**]) × 100%. TON = [**1-18a**]/[immobilized r*Aae*UPO]. N.D. = not determined. The *ee* value was determined by chiral HPLC after the derivatization of the product.

## Table 1 | Optimization of the envisioned enzymatic hemiacetal synthesis

| Entry | Immobilized r*Aae*UPO [mg] | Enzyme, [µM] | H$_2$O$_2$ [mM h$^{-1}$] | Initial rate [mM h$^{-1}$] | 1a [mM] | 1b [mM] | Selectivity [%] | TON$_{rAaeUPO}$ 1a |
|---|---|---|---|---|---|---|---|---|
| 1 | 70 | 1.02 | 6 | 2.87 | 24.8 | 4.2 | 86 | 24310 |
| 2 | 100 | 1.45 | 6 | 3.06 | 39.8 | 3.4 | 92 | 27380 |
| 3 | 125 | 1.81 | 6 | 1.02 | 24.6 | 2.8 | 90 | 13590 |
| 4 | 100 | 1.45 | 3 | 1.75 | 34.3 | 3.6 | 91 | 23660 |
| 5 | 100 | 1.45 | 9 | 5.03 | 31.5 | 2.8 | 92 | 21720 |
| 6 | 100 | 1.45 | 3 (ᵗBuOOH) | 1.96 | 8.6 | 3.6 | 70 | 5930 |
| 7 | 100 | 1.45 | 6 (ᵗBuOOH) | 2.23 | 9.8 | 8.4 | 54 | 6760 |
| 8 | 100 | 1.45 | 0 | - | 0 | 0 | - | - |
| 9 | 0 | 0 | 6 | - | 0 | 0 | - | - |
| 10 | 100 (resin only) | 0 | 6 | - | 0 | 0 | - | - |

Reaction conditions: [THF] = 0.5 mL, [immobilized r*Aae*UPO] = 70 – 125 mg (corresponding to 1.02–1.81 µM), [H$_2$O$_2$] = 3 – 9 mM h$^{-1}$, [ᵗBuOOH] = 3 – 6 mM h$^{-1}$, 30 °C, 24 h. The initial rate is based on the concentration of **1a** at 4 h. The selectivity was determined by gas chromatography (GC). Selectivity = [**1a**]/([**1a**]+[**1b**]) × 100%, TON = [**1a**]/[immobilized r*Aae*UPO]. "−" means no reaction. The reported value is based on the mean value of two distinct experiments (*n* = 2).

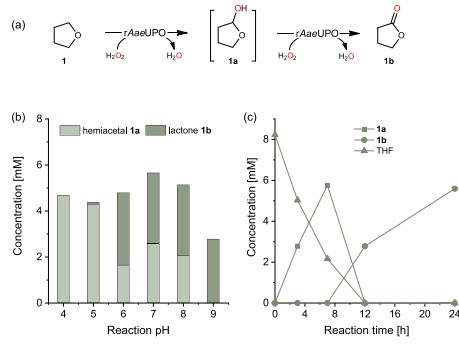

**Fig. 4 | Hydroxylation of tetrahydrofuran for γ-butyrolactone synthesis by rAaeUPO. a** Reaction scheme. **b** Influence of the reaction pH on the product distribution of hemiacetal **1a** (light green rectangles) and lactone **1b** (dark green rectangles). **c** The representative time course showing THF conversion (triangles, **1**), and tetrahydrofuran-2-ol (squares, **1a**) and γ-butyrolactone (circles, **1b**) formation. Reaction conditions for (**b**), [THF] = 8 mM, [rAaeUPO] = 1.5 μM, citrate buffer (50 mM, pH 4 − 5), or NaPi buffer (50 mM, pH 6 − 8), or Tris-HCl buffer (50 mM, pH 9), [H₂O₂] = 1 mM h⁻¹, 30 °C, 12 h. Reaction conditions for (**c**), [THF] = 10 mM, [rAaeUPO] = 1.5 μM, Tris-HCl buffer (50 mM, pH 9), [H₂O₂] = 1 mM h⁻¹, 30 °C, 24 h. The reported value is based on the mean value of two distinct experiments ($n$ = 2).

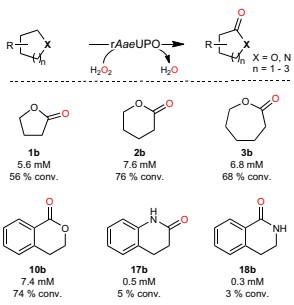

**Fig. 5 | Substrate scope of the cyclic ether oxidation reaction for lactone and lactam synthesis.** Conditions: [substrates] = 10 mM, Tris-HCl buffer (50 mM, pH 9), 5% dimethyl sulfoxide (DMSO) in **10**, **17** and **18**, [rAaeUPO] = 1.5 μM, [H₂O₂] = 1 mM h⁻¹, 30 °C, 24 h. The conversion was determined by GC. Conversion = [1-18b]/[1-18] × 100%. The reported value is based on the mean value of two distinct experiments ($n$ = 2).

from acidic to neutral and alkaline conditions remarkably influenced the formation of lactone **1b**. Below pH 5, only hemiacetal was obtained, while at pH 9 the lactone product formed exclusively. Figure 4c shows the reaction course starting from the oxidation of THF catalyzed by rAaeUPO in the presence of H₂O₂. Clearly, with the consumption of THF, hemiacetal **1a** formed first, and then a stepwise oxidation of the intermediate **1a** into lactone **1b** occurred. 5.6 mM of **1b** was obtained, corresponding to a conversion of 56% and a TON of 3733 for the enzyme. A gap in the mass balance (ca. 20%) is attributed to the hydrolysis of hemiacetal **1a** into 4-hydroxybutanal.

After the successful control of the product distribution, we selected the substrates to show rAaeUPO-catalyzed C-H bond oxyfunctionalization for lactone synthesis via the formed hemiacetal as an intermediate (Fig. 5). Lactones **1b**, **2b**, **3b**, and **10b** were formed with good conversion (56-76%). It is also possible to synthesize lactams (e.g., **17b** and **18b**) via hemiaminal, whereas the poor enzyme activity requires further improvement.

### Quantum chemical study on the mechanism and regioselectivity
While we were delighted to discover that UPO can facilitate the synthesis of cyclic hemiacetals, a reaction that has not yet been convincingly demonstrated by enzymes, however, two questions remain unanswered. The first concerns the detailed mechanism of hemiacetal

formation under neat conditions, which contradicts the established ether cleavage mechanism under aqueous conditions. The second involves determining the exact mechanism for controlling the selectivity of both hemiacetal and lactone synthesis. To address these questions, molecular dynamics (MD) simulations and quantum chemical (QC) calculations were performed in the present study (Fig. 6a, b).

The QC calculations showed that the hydroxylation of tetrahydrofuran (**1**) to form tetrahydrofuran-2-ol (**1a**) follows the typical mechanism proposed for heme-dependent enzymes and the doublet spin state is the ground state of the reaction[44–46]. After the activation of the iron porphyrin by hydrogen peroxide to form the compound I (Cpd I), the reaction proceeds with two steps consisting of the rate-limiting hydrogen atom transfer (HAT) process producing a free radical intermediate and the following low-barrier hydroxyl rebound process. Importantly, the experimentally observed regioselectivity is reproduced by QC calculations. The energy barriers for the HAT from the α-carbon and β-carbon of **1** are 10.6 kcal mol⁻¹ (**TS1-A**, Fig. 6c) and 13.1 kcal mol⁻¹ (**TS1'-A**, Fig. 6c), respectively. Thus, the barrier for the HAT of the H on the α-carbon to give tetrahydrofuran-2-ol is 2.5 kcal mol⁻¹ lower than that of the H on the β-carbon to generate tetrahydrofuran-3-ol. Analysis on the electronic structures of the transition states showed that the spin density of the α-carbon is −**0.18** at **TS1-A**, while the spin density of the β-carbon is −**0.42** at **TS1'-A**. The free radical forming in the HAT process is thus more delocalized in the former case, resulting in a lower barrier. This explanation can be generalized to all the considered substrates shown in Fig. 3. In the case of substrate **10**, it presents a unique scenario with two α-carbons. When the HAT process takes place on the α-carbon atom between the carbonyl group and the aromatic ring (C1), electrons of the radical can be more delocalized because of the proximity to the π-system of the aromatic ring, as compared to when the radical forms at the other α-carbon atom (C3). This causes hydroxylation at the C1 position more favorable over the C3 position, resulting in the formation of product **10a**.

In the step of the oxidization of **1a** to form γ-butyrolactone (**1b**), interestingly the free radical intermediate after the HAT process is unstable and a proton transfer from the hydroxyl group of **1a** to Glu196 takes place spontaneously during the optimization (Fig. 6d). The latter process is accompanied by an electron transfer from the substrate to the iron center. The following proton transfer from Glu196 to Fe-OH is highly exothermic.

Overall, the QC calculations indicate that the conversion of **1** to **1a** consists of the rate-limiting HAT process forming a free radical and the subsequent hydroxyl rebound. For the over oxidation of **1a** to **1b**, the HAT process is concerted by a proton transfer from **1a** to Glu196 due to the instability of the free radical intermediate created by the HAT. Furthermore, the regioselectivity observed in the conversion of **1** to **1a** was reproduced by the calculations as it was shown that the selectivity is resulted from the enhanced delocalization of the radical at the α-carbon of the substrate in the formation of the free radical intermediate. Additionally, the superposition of all the optimized structures of the intermediates and transition states involved in the reaction shows that the residues in the rAaeUPO active site change very little during the entire catalytic cycle (Supplementary Fig. 40).

### Umbrella sampling simulations on the pH-dependence
As discussed, an interesting finding is that the product selectivity is altered at different pH values. At pH = 4, **1a** is the only product of the rAaeUPO-catalyzed oxyfunctionalization of tetrahydrofuran, while at pH = 9 the accumulated **1a** is overoxidized to **1b** (Fig. 5). To rationalize the reasons for this product selectivity, umbrella sampling simulations were performed for the binding of **1** and **1a** to the active site of the enzyme at varied pH conditions (pH = 4 and 9) by modeling the ionizable residues in different protonation states (Fig. 7). The obtained

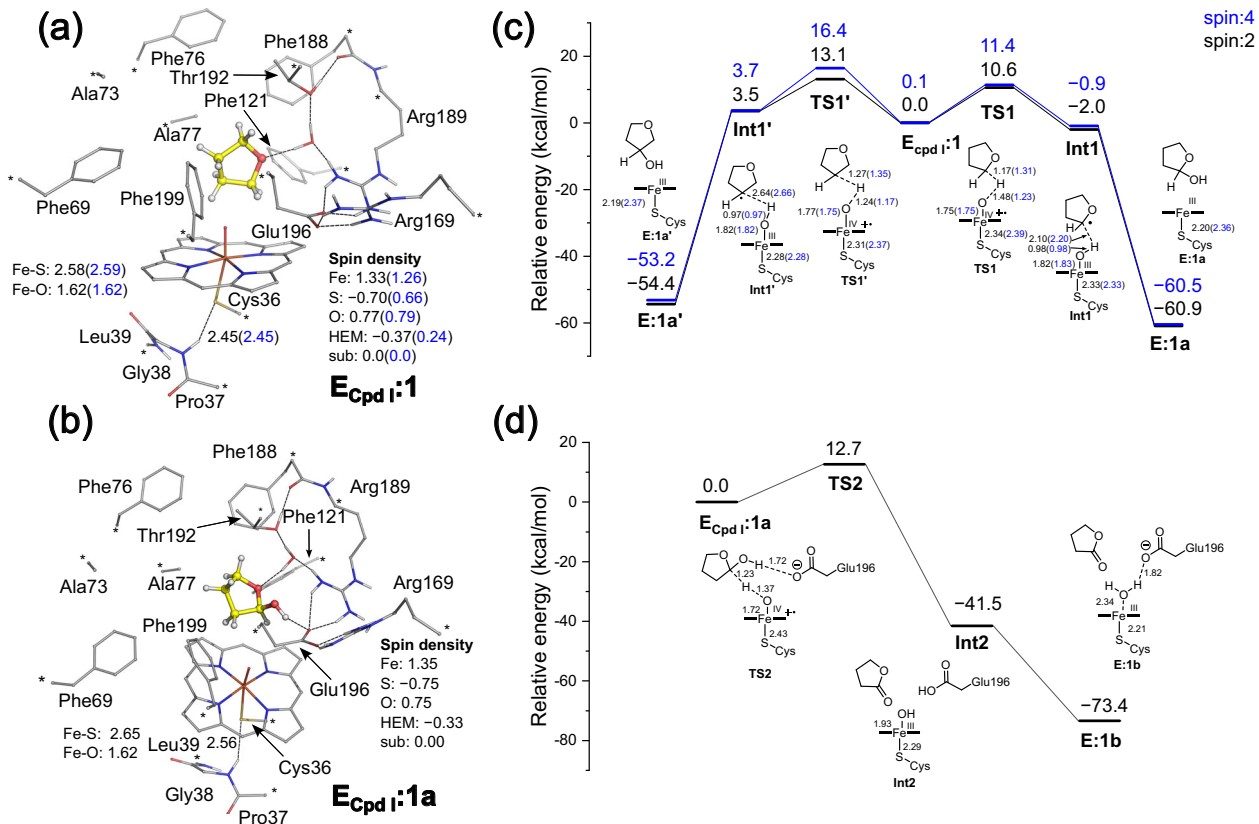

**Fig. 6 | Optimized structures and calculated energy profiles of r*Aae*UPO-catalyzed reaction.** Optimized structures of the enzyme-substrate complexes with tetrahydrofuran **1** (**a**) and tetrahydrofuran-2-ol **1a** (**b**). The "*" represents the atoms fixed in the geometry optimization. Nonpolar hydrogens of the protein and heme are hidden for clarity. Calculated energy profiles for the hydroxylation of tetrahydrofuran **1** to tetrahydrofuran-2-ol **1a** and tetrahydrofuran-3-ol **1a'** (**c**) and the oxidation of tetrahydrofuran-2-ol **1a** to γ-butyrolactone **1b** (**d**).

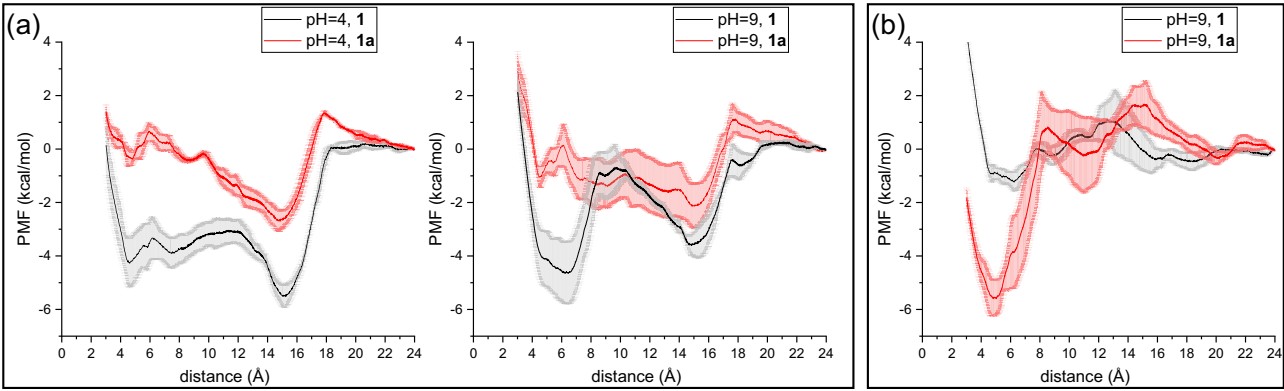

**Fig. 7 | Potential of mean force of 1 and 1a entering the active site under different condition. a** PMF of tetrahydrofuran **1** and tetrahydrofuran-2-ol **1a** entering the active site at pH = 4 (left) and pH = 9 (right). The umbrella sampling simulations were performed using the model with different protonation states of the ionizable residues in the enzyme. **b** PMF of tetrahydrofuran-2-ol **1a** entering the active site at pH = 9 in the presence of buffer molecules in the solution. The curve shows the average of values of three independent PMFs, while the error bar shows the error deviation.

potential of mean force (PMF) profiles shows that under both conditions the binding of **1** to the active site is much more favored than **1a**. Moreover, two minima were identified on the PMF profiles for all considered cases at the distances between the oxygen atom of Cpd I and the oxygen of the substrate five-membered ring with values of ca 5 Å and ca 15 Å, which can be denoted as the substrate binding inside and outside of the active site, respectively. At lower pH, **1** displays almost the same probability for binding inside and outside, but **1a** is prefers to bind outside (Fig. 7a). This explains why the conversion of **1a**

to **1b** was not experimentally detected at pH 4. When the pH increases to 9, the binding preference inside the active site is much more preferred than that outside for **1**, and the binding inside and outside are now both possible for **1a** (Fig. 7a). Therefore, at pH = 9, the conversion of **1** to **1a** easily occurs, and the oxidation of **1a** to **1b** can take place when **1a** accumulates to a certain amount. This is highly consistent with the experimental results. The impact of phosphate buffer molecules at pH = 9 on the binding affinities of the substrate was also explored (Fig. 7b), and the calculations show that the presence of

buffer molecules significantly enhances the binding of **1a** to the active site, resulting in **1a** being much more preferentially located in the active site. Thus, in light of the current simulations the differences in the catalytic performance of the r*Aae*UPO reaction between pH 4 and pH 9 can be attributed to the distinct binding preferences of **1** and **1a** to the active site of the enzyme under the two pH conditions.

In summary, this work used the strategy of reaction engineering and demonstrated that unspecific peroxygenase from *Agrocybe aegerita* was capable of synthesizing optically pure cyclic hemiacetal compounds through the selective C-H bond oxyfunctionalization of cyclic ethers. Under neat reaction conditions, hydrolysis of the hemiacetal products was remarkably circumvented, leading to a unexplored application of peroxygenases compared to the established demethylation reactions. A diverse range of substrates was converted into the corresponding hemiacetal products with up to >99% *ee*. Moreover, by switching the reaction conditions to aqueous conditions the hemiacetal products were further oxidized into lactones, which represents an in situ application of the hemiacetals. The formation of lactam products was also possible, however, the low enzyme activity needs to be addressed. In virtue of molecular dynamics simulations and quantum chemical calculations, we proposed reaction mechanisms for the peroxygenase-catalyzed conversion of cyclic ether to hemiacetal, and subsequently to lactone. The differences in the distribution of hemiacetal and lactone products can be explained by the binding preferences of the cyclic ether and hemiacetal to the enzyme's active site. This work demonstrates an enzymatic method for hemiacetal synthesis and highlights the potential of peroxygenase as a versatile biocatalyst in synthetic chemistry.

## Methods

### Materials

Unless otherwise stated, all reagents and solvents were purchased from commercial suppliers (J&K Chemical, Bide Pharmatech Ltd., Macklin, Energy Chemical, etc.) and used without purification. Anhydrous toluene and dichloromethane were obtained by distilling the solvent using calcium hydride and stored under nitrogen. The resin carriers were purchased from Mreda, Macklin and Sunresin New Materials Co. Ltd. and used as described in the immobilization of r*Aae*UPO part of the Supplementary Information.

### Enzyme preparation

The procedures of the recombinant expression and purification of the evolved nonspecific peroxygenase mutant (PaDa-I) from *A. aegerita* in *P. pastoris* were based on reported procedures[29]. Typically, the single colony from the *P. pastoris* clones were picked and inoculated in BMGY medium (5 mL). After incubation for 36 h at 30 °C, 1 mL of the preculture was transferred into 100 mL of BMGY medium and incubated for 24 h till $OD_{600}$ reached around 1.5. Methanol was then added and the incubation was continued for 72 h. After the fermentation process, the culture broth of *P. pastoris* cells containing r*Aae*UPO was separated by centrifugation (10956 × g, 4 °C, 1 h). The supernatant was filtered through a 0.22 μm filter and kept at −80 °C for later use.

### Typical procedures for hemiacetal synthesis

To a 1 mL transparent glass vial, 100 mg (corresponding to 1.45 μM) of immobilized r*Aae*UPO and 0.5 mL of substrate were added. Then, $H_2O_2$ from a stock solution (300 mM) was injected by a syringe pump into the mixture at rate of 6 mM h$^{-1}$ (10 μL h$^{-1}$). The reaction vial was sealed and agitated in a thermal shaker at 30 °C and 800 rpm. At intervals, aliquots were withdrawn, 20 μL of the mixture was mixed with 180 μL of ethyl acetate (containing 5 mM of dodecane as an internal standard). Then, the sample was dried over $Na_2SO_4$ and analyzed by gas chromatography (GC) to determine the product concentration. To measure the optical purity, HPLC was used after the product was derivatized by 2-methylbenzoyl chloride (acetic anhydride for **12a**).

Details of the GC and HPLC methods are shown in Supplementary Tables 4, 5 and Supplementary Figs. 6–18, 28–36. For the detailed experimental procedures and analytics of the hemiacetal synthesis, see Supplementary Information.

### Computational study

To shed light on the reaction mechanism and regioselectivity of r*Aae*UPO-catalyzed oxyfunctionalization of the C-H bond of tetrahydrofuran, molecular dynamics (MD) simulations and quantum chemical (QC) calculations were performed on the basis of the previously solved crystal structure[47]. Optimized geometries and spin densities of the transition states and intermediates calculated by QC are shown in Supplementary Figs. 37–39. The root mean square deviation (RMSD) and distribution histograms of the MD simulations are shown in Supplementary Figs. 40, 41. Details of the methods are included in the Supplementary Information.

### Reporting summary

Further information on research design is available in the Nature Portfolio Reporting Summary linked to this article.

## Data availability

All data generated in this study are provided in the Supplementary Information/Source Data file. The source data underlying Table 1, Figs. 2a, b, 4a, b, 7a, b, and Supplementary Figs. 2–5 are provided as a source data file. PDB file used in this study is available in Protein Data Bank (PDB) (ID: 6ekz, https://doi.org/10.2210/pdb6ekz/pdb). These data are also available from the corresponding authors upon request. Source data are provided with this paper.

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

## Acknowledgements

The authors acknowledge financial support from the National Key R&D Program of China (2023YFC3403600 to W.Z.), the National Natural Science Foundation of China (No. 32171253 to W.Z. and No. 22103095 to X.S.) and Tianjin Synthetic Biotechnology Innovation Capacity Improvement Project (No. TSBI-CIP-CXRC-032 to W.Z. and No. TSBICIP-CXRC-026 to X.S.).

## Author contributions

X.H., H.L., R.G., and Q.S. performed the experimental work and analyzed the results; F.C. and X.S. performed the computational study and analyzed the results; P.D. participated in the analytics and validation of the experimental results; W.Z. conceived and designed the experiments, W.Z. and X.S. co-wrote the manuscript. All authors participated in the writing of the manuscript.

## Competing interests
The authors declare no competing interests.
