## [Peer Review File · Nature Communications]

REVIEWER COMMENTS

Reviewer #1 (Remarks to the Author):

In this work, authors used immobilized peroxygenase from *Agrocybe aegerita* (AaeUPO) under regular reaction conditions to allow for the selective C-H bond oxyfunctionalization of environmentally significant cyclic ethers to cyclic hemiacetals. A wide range of cyclic ethers with diverse structures were converted into the corresponding hemiacetals. The main novelty of this work is the use of neat reaction conditions to avoid the instability of hemiacetals in aqueous environments. This strategy is useful for the synthesis of various hemiacetals and also for other similar compounds unstable in water. Authors can consider the following comments to improve the manuscript.

1. One major drawback of this work is the stereo-selectivity. The authors did not provide any data of stereoselectivity for this peroxygenase-catalyzed oxidation, although high stereoselectivity is one of the most remarkable advantages of enzyme.
2. In Figure 2 and 4, although authors provided the product concentrations, it is difficult to evaluate these reactions without the information of starting concentrations of substrates in these figures. It is better to provide the conversions or yields of these substrates.
3. Why the efficiency of immobilized rAaeUPO in using H₂O₂ is significantly higher than that of tBuOOH in terms of product concentration and selectivity. Can the author provide an explanation?
4. For the detailed mechanism of hemiacetal formation under neat conditions and the exact mechanism for controlling the selectivity of both hemiacetal and lactone synthesis, a simple summary should be made respectively, rather than stacking the results together.
5. Line 207, the discussion about the regioselectivity is not clear, how to get 2.5 kcal/mol energy difference from Figure 5c? Moreover, concerning the regioselectivity of hydroxylation, it is also interesting to provide some discussion about the substrate 10a, where the hydroxylation has two potential positions with similar characteristics.
6. Please check the whole manuscript carefully to correct some errors, such as Line 72: add (b) in the figure caption; Line 127, there is no entry 10 in Table 1.

Reviewer #2 (Remarks to the Author):

The authors present a study of the application of a peroxygenase enzyme to water sensitive substrates at high concentration. The work is an impressive addition to the previous work by Zhang, in which the capabilities of UPOs have been greatly enhanced by the use of immobilised enzyme in what is essentially pure substrate.

- What are the noteworthy results?

The noteworthy result is the application of the immobilised UPO to the oxidation of ethers to hemiacetals in a concentration intensive manner – these reactions would be unthinkable with, for example, P450 catalysts.

- Will the work be of significance to the field and related fields? How does it compare to the established literature? If the work is not original, please provide relevant references.

Yes – very significant, also when included with the authors' work on for example, toluene oxidation also using enzyme in pure substrate.

- Does the work support the conclusions and claims, or is additional evidence needed?

Yes – the work needs some comment on the enantioselectivity of hydroxylation of the substrates and optical purity of the products.

- Are there any flaws in the data analysis, interpretation and conclusions? - Do these prohibit publication or require revision?

Not from this perspective; most data appear to be in order

- Is the methodology sound? Does the work meet the expected standards in your field?

- Is there enough detail provided in the methods for the work to be reproduced?

Yes

Some minor comments:

Abstract Line 18 'was converted'

Abstract Line 19 'of up to 95100' Is this TTNs?

Intro Page 2 Line 34 'embedded'

Intro Page 3 line 55 –could lose this line ‘The above mentioned..’

Intro Page 3 para 2 – Should use ‘UPO’ instead of ‘peroxygenase’ throughout this paragraph (and the whole paper) as the abbreviation has been introduced by the author, and helps to distinguish UPOs from, for example P450 peroxygenases.

Results Page 4 line 80 ‘We began by evaluating..using the UPO..’

Results Page 4 lines 83-84 – presumably the PaDa-I variant? This should be made clear

Results Page 4 Line 95 ‘peroxygenase, in which only..’

Results Page 5 Line 109-110 ‘100 mg’ is not a concentration; please clarify

Results Page 6 Line 135 ‘electronic’

Results Page 8 – Why is there no mention of enantioselectivity for the results in Figure 2. Was there none? This is worth reporting. Even alpha-D measurements of 0 deg would be informative.

Page 8 Line 159 ‘we wondered whether’

Page 10 Line 184 – remove lines 184 – 185 - this appears to be just speculation; Why would this particular UPO be any better?

Page 10 Line 194 ‘enzymes, however two questions..’

Page 12 – Speculations on the binding preferences of substrates to the UPO would be better supported by crystal complexes of this UPO with the relevant substrates

Reviewer #3 (Remarks to the Author):

I was invited to evaluate the quantum chemical and molecular dynamics part of the study.

The authors present two computational results:

- They calculated the energy profiles for the reaction of the hydroxylation of tetrahydrofuran and the oxidation of the main product at low pH to butyrolactone. They used the QM cluster model method at the level of Density Functional Theory (B3LYP-D3(BJ)/def2-SVP) including solvent effects, and calculated zero-point correct energies.

- They performed classical molecular dynamics (MD) simulations, steered MD simulations, and umbrella sampling to explain the pH-dependent product selectivity of the enzyme.

Both approaches seem to me to be adequate to answer the scientific questions posed by the authors. However, I have the following concerns:

1. The calculation methods of the supporting data need to be described in more detail. In particular, the protocol for all steps of the classical MD simulations and the umbrella sampling protocols need lack of important information: algorithms used for the minimization, how the system is constrained during the heating step (value of the harmonic force constant, which atoms?), how did the author attach the substrate to the system before running the steered MD simulation, which algorithm did the authors use to cluster the MD trajectory and how many clusters did they compute, and more importantly, for how much time was the system simulated in the production phase?

2. Given the small differences for the energy barriers of TS1 and TS1* in Figure 5, the authors need to justify why they chose B3LYP-D3(BJ)/def2-SVP to study the catalysed reaction mechanism. Staying at the level of DFT, different DFT functionals will yield different energy barriers. In addition to that, it is known that DFT overstabilizes delocalized states, most intense in systems with an odd number of electrons (e.g., see 10.1063/1.1926277). The authors should comment on that and again, justify why they decided to use B3LYP (+ basis set) and not other DFT functionals. The agreement of B3LYP with some experimental trend can be caused by chance or self-cancellation error. It is curious that the author did not mention any theoretical work with heme-proteins or peroxidases from the literature although they claim that the 'hydroxylation of tetrahydrofuran (...) follows the typical mechanism proposed for heme-dependent enzymes and doublet spin state is the ground state of the reaction'. The authors need to solve this. In addition to that, have the author try any wave function method?

3. What is the error/standard deviation in the PMF profile obtained using umbrella sampling? (see Fig. 6).

4. Finally, what happens to the dynamics of the enzyme at the active site during the catalytic cycle? Do the side chains move? Please comment on this.

Comments	Answers
Reviewer 1	
Reviewer #1 (Remarks to the Author): In this work, authors used immobilized peroxygenase from Agrocybe aegerita (AaeUPO) under regular reaction conditions to allow for the selective C-H bond oxyfunctionalization of environmentally significant cyclic ethers to cyclic hemiacetals. A wide range of cyclic ethers with diverse structures were converted into the corresponding hemiacetals. The main novelty of this work is the use of neat reaction conditions to avoid the instability of hemiacetals in aqueous environments. This strategy is useful for the synthesis of various hemiacetals and also for other similar compounds unstable in water. Authors can consider the following comments to improve the manuscript.	We would like to thank the reviewer for sharing the enthusiasm with us on a novel strategy to enable an enzymatic synthesis of hemiacetals! We believe that this work will be of immediate interest to the community of biocatalysis, enzymology and synthetic chemistry.
1. One major drawback of this work is the stereo-selectivity. The authors did not provide any data of stereoselectivity for this peroxygenase-catalyzed oxidation, although high stereoselectivity is one of the most remarkable advantages of enzyme.	We thank the reviewer for this very important comment and fully agree with him/her that the stereoselectivity will build up the novelty significantly. Pleasantly, we have solved the analytics after some effort. The results on the stereoselectivity have been provided in the revised manuscript, which demonstrated that the current approach allowed excellent stereoselectivity in the hemiacetal synthesis (up to >99% ee in R-configuration).
2. In Figure 2 and 4, although authors provided the product concentrations, it is difficult to evaluate these reactions without the information of starting concentrations of substrates in these figures. It is better to provide the conversions or yields of these substrates.	We are most grateful for this comment! Indeed, the reviewer is right mentioning that the conversions attainable in the solvent-free system are not satisfactory. However, we would like to highlight the role of the neat reaction condition: the substrate also serves a dual role: the substrate itself and the solvent (substrate and product sink). In the latter case the solvent enabled the stability of both the enzyme and the hemiacetal product. Following the comment here, we have re-thought our approach aiming at higher

	conversions. The first strategy is simply to increase the catalyst loading as shown below, in which the product concentration double when using tetrahydropyran as substrate.  The second strategy is to use an inert solvent such as acetone. Pleasingly, acetone as solvent proofed to be successful thereby also eliminating the need for further extraction. In this reaction 200 mM of the tetrahydropyran was completely converted after 40 h.  However, we were not sure to include the above results to the current manuscript as it can change the story line. In this case, we prefer to use the pure acetone as solvent for a coming synthetic manuscript. We hope that the reviewer will agree with us.
3. Why the efficiency of immobilized rAeUPO in using H₂O₂ is significantly higher than that of tBuOOH in terms of product concentration and selectivity. Can the author provide an explanation?	We thank the reviewer for this important reminder. Indeed the state-of-the-art peroxygenase catalysis under neat conditions suggested that tBuOOH serves a better oxidant in comparison to the H₂O₂. However, after a thorough literature research we found that so far these reactions have been mostly performed in nonpolar solvents (such as ethyl benzene, toluene, benzene, etc.) We found a publication which showed a similar phenomenon with our study that H₂O₂ could be otherwise a better oxidant in a rather polar

	solvent. In the revised manuscript, the explanation has been added 'The kinetically less effective tBuOOH has a good affinity with the unspecific peroxygenase in apolar solvents, whereas H₂O₂ is expected to have a higher affinity with the enzyme in slightly polar solvent (i.e. THF)'.
4. For the detailed mechanism of hemiacetal formation under neat conditions and the exact mechanism for controlling the selectivity of both hemiacetal and lactone synthesis, a simple summary should be made respectively, rather than stacking the results together.	We thank the reviewer for offering his/her suggestions on the discussion part. We have now incorporated summaries concerning the reaction mechanisms and selectivity in the Discussion section: On pages 11 and 12: "Overall, the QC calculations indicate that the conversion of 1 to 1a consists of the rate-limiting HAT process forming a free radical and the subsequent hydroxyl rebound. For the oxidation of 1a to 1b, the HAT process is concerted by the proton transfer from 1a to Glu196 due to the instability of the free radical intermediate created by the HAT. Furthermore, the regioselectivity observed in the conversion of 1 to 1a was reproduced by the calculations and it was shown that the selectivity is resulted by the enhanced delocalization of the radical at the α-carbon of the substrate in the formation of the free radical intermediate." On pages 13: "In light of the current simulations, the differences in the catalytic performance of the rAaeUPO reaction between pH 4 and pH 9 can be attributed to the distinct binding preferences of 1 and 1a to the active site of the enzyme under the two pH conditions." Additionally, a short summary has also been added to the Conclusion: "In virtue of molecular dynamics simulations and quantum chemical calculations, we proposed reaction mechanisms for the rAaeUPO-catalyzed conversion of cyclic ether to hemiacetal, and subsequently to lactone. The differences in the distribution of hemiacetal and lactone products can be explained by the different binding preferences of the cyclic ether and

	hemiacetal to the enzyme's active site. "
5. Line 207, the discussion about the regioselectivity is not clear, how to get 2.5 kcal/mol energy difference from Figure 5c? Moreover, concerning the regioselectivity of hydroxylation, it is also interesting to provide some discussion about the substrate 10a, where the hydroxylation has two potential positions with similar characteristics.	We are sorry for this confusion. The original text has now been modified to: "Importantly, the experimentally observed regio-selectivity is reproduced by the QC calculations. The energy barriers for the HAT from the α-carbon and β-carbon are 10.6 kcal/mol (TS1-A, Figure 5c) and 13.1 kcal/mol (TS1'-A, Figure 5c), respectively. Thus, the barrier for the HAT of the H on the α-carbon to give tetrahydrofuran-2-ol is 2.5 kcal/mol lower than that of the H on the β-carbon to generate tetrahydrofuran-3-ol." Analysis on the regio-selectivity of the substrate 10 has been added in the revised manuscript: "This explanation can be generalized to all the considered substrates shown in Figure 2. In the case of substrate 10, it presents a unique scenario with two α-carbons. When the HAT process takes place on the α-carbon atom between the carbonyl group and the aromatic ring (C1), electrons of the radical can be more delocalized because of the proximity to the π-system of the aromatic ring, compared to when the radical forms at the other α-carbon atom (C3). This causes hydroxylation at the C1 position to be preferred over the C3 position, resulting in the formation of product 10a."
6. Please check the whole manuscript carefully to correct some errors, such as Line 72: add (b) in the figure caption; Line 127, there is no entry 10 in Table 1.	We would like to thank the reviewer again for his/her effort in helping us improve our manuscript. The manuscript has now been completely revised.
Reviewer 2	
Reviewer #2 (Remarks to the Author): The authors present a study of the application of a peroxygenase enzyme to water sensitive substrates at high concentration. The work is an impressive addition to the previous work by Zhang, in which the capabilities of UPOs have been greatly enhanced by the use of immobilised enzyme in what is essentially pure substrate.	We would like to thank the reviewer for sharing his/her enthusiasm with us on a novel enzymatic strategy for the synthesis of water-sensitive hemiacetals! This work expands the reaction scopes of unspecific peroxygenase. We believe that this work will be of immediate interest to the community of biocatalysis, enzymology and synthetic chemistry.
- What are the noteworthy results? The noteworthy result is the application of the immobilised UPO to the oxidation of ethers to hemiacetals in a concentration intensive manner – these reactions would be unthinkable with, for example, P450 catalysts.	We thank the reviewer for recognizing the strength of the envisioned approach in the field of biocatalysis.
- Will the work be of significance to the field	We thank the reviewer again for recognizing

and related fields? How does it compare to the established literature? If the work is not original, please provide relevant references. Yes – very significant, also when included with the authors' work on for example, toluene oxidation also using enzyme in pure substrate.	the value of the envisioned enzymatic approach for organic synthesis.
- Does the work support the conclusions and claims, or is additional evidence needed? Yes – the work needs some comment on the enantioselectivity of hydroxylation of the substrates and optical purity of the products.	We thank the reviewer for this very important comment and fully agree with him/her that the stereoselectivity will build up the novelty significantly. Pleasantly, we have solved the analytics after some effort. The results on the stereoselectivity have been provided in the revised manuscript, which demonstrated that the current approach allowed excellent stereoselectivity in the hemiacetal synthesis (up to >99% ee in R-configuration).
- Are there any flaws in the data analysis, interpretation and conclusions? - Do these prohibit publication or require revision? Not from this perspective; most data appear to be in order	We have completely revised our manuscript based on all reviewers' comments. The corrections were made on experimental section, data analysis, interpretation and discussions.
- Is the methodology sound? Does the work meet the expected standards in your field?	We have completely revised our manuscript based on all reviewers' comments. The corrections were made on experimental section, data analysis, interpretation and discussions. Hopefully this revision will satisfy the reviewers.
- Is there enough detail provided in the methods for the work to be reproduced? Yes	We have completely revised our manuscript based on all reviewers' comments. The corrections were made on experimental section, data analysis, interpretation and discussions.
Some minor comments: Abstract Line 18 'was converted'	Corrected as requested. Moreover, the whole manuscript has been corrected carefully.
Abstract Line 19 'of up to 95100' Is this TTNs?	Yes, here we meant turnover number as mentioned in the original text, which was defined as following: TON = moles of product divided by moles of catalyst used.
Intro Page 2 Line 34 'embedded'	Corrected as requested.
Intro Page 3 line 55 –could lose this line 'The above mentioned..'	Thank you for your suggestions. Proper corrections have been made.
Intro Page 3 para 2 – Should use 'UPO' instead of 'peroxygenase' throughout this paragraph (and the whole paper) as the abbreviation has	We thank the reviewer for this detailed comment and fully agree with him/her. The whole text has been checked and corrected.

been introduced by the author, and helps to distinguish UPOs from, for example P450 peroxygenases.	
Results Page 4 line 80 'We began by evaluating..using the UPO..'	Corrected as suggested.
Results Page 4 lines 83-84 – presumably the PaDa-I variant? This should be made clear	We are sorry for this confusion. The information on the PaDa-I variant has been added to make it clear.
Results Page 4 Line 95 'peroxygenase, in which only..'	Corrected as suggested.
Results Page 5 Line 109-110 '100 mg' is not a concentration; please clarify	Corrected as suggested.
Results Page 6 Line 135 'electronic'	Corrected as suggested.
Results Page 8 – Why is there no mention of enantioselectivity for the results in Figure 2. Was there none? This is worth reporting. Even alpha-D measurements of 0 deg would be informative.	We appreciate this valuable comment! We have now solved the challenges in the analytics for determining the optical purity. Most of the products showed >99% ee in R-configuration, the results have been included in the revised version.
Page 8 Line 159 'we wondered whether'	Corrected as suggested.
Page 10 Line 184 – remove lines 184 –185 - this appears to be just speculation; Why would this particular UPO be any better?	Removed as suggested. We appreciate the suggestions here!
Page 10 Line 194 'enzymes, however two questions..'	Corrected as suggested.
Page 12 – Speculations on the binding preferences of substrates to the UPO would be better supported by crystal complexes of this UPO with the relevant substrates	We fully agree with the reviewer that a crystal structure of the complex will add extra value on the binding preference to the substrate to the enzyme pocket. After managing the optical measurement in the revised manuscript, we believe that the current results, along with the computational study, provide sufficient information on the substrate binding a catalytic cycle for the chiral hemiacetal synthesis.
Reviewer 3	
Reviewer #3 (Remarks to the Author): I was invited to evaluate the quantum chemical and molecular dynamics part of the study. The authors present two computational results: - They calculated the energy profiles for the reaction of the hydroxylation of tetrahydrofuran and the oxidation of the	We thank the referee for sharing his/her expertise in improving our manuscript.

main product at low pH to butyrolactone. They used the QM cluster model method at the level of Density Functional Theory (B3LYP-D3(BJ)/def2-SVP) including solvent effects, and calculated zero-point correct energies. - They performed classical molecular dynamics (MD) simulations, steered MD simulations, and umbrella sampling to explain the pH-dependent product selectivity of the enzyme.	
Both approaches seem to me to be adequate to answer the scientific questions posed by the authors. However, I have the following concerns:	Following the comments and suggestions here, we have now carefully revised the computational part, hopefully the changes will satisfy the reviewer.
1. The calculation methods of the supporting data need to be described in more detail. In particular, the protocol for all steps of the classical MD simulations and the umbrella sampling protocols need lack of important information: algorithms used for the minimization, how the system is constrained during the heating step (value of the harmonic force constant, which atoms?), how did the author attach the substrate to the system before running the steered MD simulation, which algorithm did the authors use to cluster the MD trajectory and how many clusters did they compute, and more importantly, for how much time was the system simulated in the production phase?	Thank you for your detailed comments. We have incorporated more detailed information into the supporting data according to your suggestion. First, the calculation details of the classic MD simulation are supplemented as follows: ‘...First, under the constraint of using harmonic force of 15 kcal/mol/A² on the protein and substrate, the solvent molecules are minimized. During this process, the steepest descent method is used to optimize for 2000 steps, followed by the conjugate gradient method for another 2000 steps. After the minimization of the solvent molecules is completed, the protein constraints are released to minimize the entire system. The steepest descent method is employed again for 5000 steps, followed by the conjugate gradient method for another 5000 steps. Subsequently, the system is gradually heated under NVT ensemble, starting from 0 K and increasing to 300 K using a time step of 1 fs for a heating duration of 100 ps. The heating process imposes restrictions on the protein and substrate with a harmonic force constant of 15 kcal/mol/A². Then, at a target temperature of 300 K and a pressure of 1.0 atm, the density of the system is equilibrated for 500 ps under the NPT ensemble, and a harmonic force of 5 kcal/mol/A² was used to constrain the position

	of the substrate with Cpd I in the process. The system followed by a productive MD run of 100ns.' Secondly, the calculation details of the Umbrella sampling are supplemented as follows: '...In addition, THF and tetrahydrofuran-2-ol were manually placed at the active site. Before starting steered MD simulations, the substrate was fixed at the active site and simulated for 20ns, and the simulated structure was used as the initial structure of steered MD, the harmonic force used is 5 kcal /mol/A².' Regarding the cluster analysis, we divided the 10,000 frame conformations generated by MD simulation into 10 clusters using the hierarchical clustering method. The conformation with the largest proportion was selected to construct the quantum chemical cluster model. This part of the content has been modified to "The quantum chemical cluster model was built based on the MD simulation results. Hierarchical clustering methods was conducted on the 10,000 frames of molecular dynamics trajectories of rAaeUPO in the THF solution. The structure with the highest number of conformations was selected as the initial structure to build the QM model." in the Supporting Information.
2. Given the small differences for the energy barriers of TS1 and TS1* in Figure 5, the authors need to justify why they chose B3LYP-D3(BJ)/def2-SVP to study the catalysed reaction mechanism. Staying at the level of DFT, different DFT functionals will yield different energy barriers. In addition to that, it is known that DFT overstabilizes delocalized states, most intense in systems with an odd number of electrons (e.g., see 10.1063/1.1926277). The authors should comment on that and again, justify why the decided to use B3LYP (+ basis set) and not other DFT functionals. The agreement of B3LYP with some experimental trend can be	We did not try wave function method in the present study. Our choice to employ the density functional theory, specifically B3LYP-D3(BJ), as our functional in the present study was not arbitrary. It made on the basis of success applications of this functional on different heme-dependent enzymes in the literature, which provide reliable information on the reaction mechanisms and selectivities. We have added the following sentence to the QM Cluster Model and Methods section in the Supporting Information (page S52): "It has been established that density functional theory (DFT), in particular the B3LYP functional with the Grimme dispersion correction, offers

caused by chance or self-cancellation error. It is curious that the author did not mention any theoretical work with heme-proteins or peroxidases from the literature although they claim that the 'hydroxylation of tetrahydrofuran (...) follows the typical mechanism proposed for heme-dependent enzymes and doublet spin state is the ground state of the reaction'. The authors need to solve this. In addition to that, have the author try any wave function method?	reliable insights into the reaction mechanisms and selectivities of heme-dependent enzymes[27-30]. Therefore, B3LYP-D3(BJ) was chosen to investigate the reaction mechanism of rAaeUPO-catalyzed transformation in the present study.” Recent review papers on the computational studies on the heme-dependent enzymes are cited (references 27 to 30) in the revised manuscript. Recent review papers have been cited in the discussion of “hydroxylation of tetrahydrofuran (...) follows the typical mechanism proposed for heme-dependent enzymes and doublet spin state is the ground state of the reaction” (references 43 to 46) in the revised manuscript.
3. What is the error/standard deviation in the PMF profile obtained using umbrella sampling? (see Fig. 6).	Thank you for your comment. A figure with error deviation has now been added in the revised manuscript (Figure 6).
4. Finally, what happens to the dynamics of the enzyme at the active site during the catalytic cycle? Do the side chains move? Please comment on this.	Thank you for this very important comment. We overlapped all the optimized structures of the intermediates and transition states involved in the reaction and it was shown that the residues in the UPO active site change very little during the entire catalytic cycle. A short discussion on this has been added in the revised manuscript, and the superposition has been added to the Supporting Information (Supplementary Figure 39).   Supplementary Figure 54. Superposition of all the optimized structures of the intermediates and transition states involved in the reaction pathway of rAaeUPO-catalyzed reaction.

Editorial comments	
Important: In addition to the above, you must comply with the following editorial requests; we will not be able to proceed with your revised manuscript otherwise. Please also see the Nature Communications formatting instructions, which you may find useful while preparing your revised manuscript.	We thank the editorial office for very detailed guidance. We have revised the manuscript and SI files and made other necessary changes based on the mentioned polices.
POLICIES AND FORMS REQUIRED FOR RESUBMISSION * Please complete or update the following checklist(s) to verify compliance with our research ethics and data reporting standards. Address all points on the checklist, revising your manuscript in response to the points if needed. The form(s) must be downloaded and completed in Adobe Reader rather than opened in a web browser. Each form must be uploaded as a Related Manuscript file at the time of resubmission. Editorial policy checklist: https://www.nature.com/documents/nr-editorial-policy-checklist.pdf	The policy has been followed and the checklist has been uploaded.
Reporting summary: https://www.nature.com/documents/nr-reporting-summary.pdf	Uploaded as requested.
Your work characterises chemical or biomolecular materials. Please see the link below for reporting requirements. There is no form to upload but you may need to revise your manuscript to comply with this policy. https://www.nature.com/ncomms/submit/chemical-characterisation	All chemical structures have been redrawn based on the guidance of the nature journals.
DATA AND CODE AVAILABILITY * All Nature Communications manuscripts must include a "Data Availability" section after the Methods section but before the References. If any of the data can only be	The policy has been followed and the necessary files have prepared and uploaded.

shared on request or are subject to restrictions, please specify the reasons and explain how, when, and by whom the data can be accessed. For more information on this policy and a list of examples, see: <https://www.nature.com/documents/nr-data-availability-statements-data-citations.pdf>

* As Nature Portfolio policies strongly encourage you to share your research data in a public repository (e.g. spreadsheets, text, images), we are partnering with the figshare repository so that you can use the figshare integration via the 'Research Data Deposition' tab when submitting your revised manuscript.

Data are stored privately until a manuscript decision is reached and you can edit/withdraw them up to this point: you retain rights and control over your data. The data will be published at the same time as your article; you will receive a data DOI, with guidance on linking the data and manuscript. In the event your manuscript is not accepted, you can keep or remove your data in figshare.

We recommend the use of discipline-specific repositories where available and for a number of data types this is mandatory. Ensure you do not submit these data types or any sensitive data to figshare.

* We strongly encourage you to deposit all new data associated with the paper in a persistent repository where they can be freely and enduringly accessed. We recommend submitting the data to discipline-specific and community-recognised repositories; a list of repositories is provided here: <http://www.nature.com/sdata/policies/repositories>
Refer to our data policies here: [P
A
G
E
1
1
-](https://www.nature.com/nature-portfolio/editorial-policies/reporting-standards#availability-of-data * To maximise the reproducibility of research data, we strongly encourage you to provide a file containing the raw data underlying the following types of display items:  - Any reported means/averages in box plots, bar charts, and tables - Dot plots/scatter plots, especially when there are overlapping points - Line graphs 	
The data should be provided in a single Excel file with data for each figure/table in a separate sheet, or in multiple labelled files within a zipped folder. Name this file or folder 'Source Data', and include a brief description in your cover letter. The "Data Availability" section should also include the statement "Source data are provided with this paper." To learn more about our motivation behind this policy, please see: https://www.nature.com/articles/s41467-018-06012-8	We have now prepared the source data file and included in the information in the manuscript.
We also mandate the presentation of uncropped versions of any gels or blots, labelled with the relevant panel and identifying information such as the antibody used.	We fully understand this policy, however, this is not applicable in our case as no gel images were included in the manuscript and SI.
Please replace your bar graphs with plots that feature information about the distribution of the underlying data. All data points should be shown for plots with a sample size less than 10. For larger sample sizes, please consider box-and-whisker or violin plots as alternatives. Measures of centrality, dispersion and/or error bars should be plotted and described in the figure legend.	All figures with error bars have been replotted.
ORCID * Nature Communications is committed to improving transparency in authorship. As part of our efforts in this direction, we are now requesting that all authors identified as 'corresponding author' create and link their	ORCID has been linked to three corresponding authors.

Open Researcher and Contributor Identifier (ORCID) with their account on the Manuscript Tracking System prior to acceptance. ORCID helps the scientific community achieve unambiguous attribution of all scholarly contributions.	
--	--

REVIEWERS' COMMENTS

Reviewer #1 (Remarks to the Author):

The authors have addressed my concerns through revision of the text. I have no further comments and I recommend it for publication in Nature Communications.

Reviewer #3 (Remarks to the Author):

The authors have replied to all the questions I raised in my previous report concerning the computational part. I appreciate the changes made by the authors in the MS and SI. I have no further questions.